# Circulating Interleukin-8 Dynamics Parallels Disease Course and Is Linked to Clinical Outcomes in Severe COVID-19

**DOI:** 10.3390/v15020549

**Published:** 2023-02-16

**Authors:** Ranit D’Rozario, Deblina Raychaudhuri, Purbita Bandopadhyay, Jafar Sarif, Priyanka Mehta, Chinky Shiu Chen Liu, Bishnu Prasad Sinha, Jayasree Roy, Ritwik Bhaduri, Monidipa Das, Sanghamitra Bandyopadhyay, Shekhar Ranjan Paul, Shilpak Chatterjee, Rajesh Pandey, Yogiraj Ray, Dipyaman Ganguly

**Affiliations:** 1IICB-Translational Research Unit of Excellence, CSIR-Indian Institute of Chemical Biology, Kolkata 700091, India; 2Academy of Scientific and Innovative Research (AcSIR), Ghaziabad 201002, India; 3INtegrative GENomics of HOst-PathogEn (INGEN-HOPE) Laboratory, CSIR-Institute of Genomics and Integrative Biology, New Delhi 110007, India; 4Indian Statistical Institute, Kolkata 700108, India; 5Department of Medicine, ID & BG Hospital, Kolkata 700010, India; 6Department of Infectious Diseases, Institute of Postgraduate Medical Education and Research, Kolkata 700020, India

**Keywords:** COVID-19, cytokines, machine learning, IL-8, survival, biomarker

## Abstract

Severe COVID-19 frequently features a systemic deluge of cytokines. Circulating cytokines that can stratify risks are useful for more effective triage and management. Here, we ran a machine-learning algorithm on a dataset of 36 plasma cytokines in a cohort of severe COVID-19 to identify cytokine/s useful for describing the dynamic clinical state in multiple regression analysis. We performed RNA-sequencing of circulating blood cells collected at different time-points. From a Bayesian Information Criterion analysis, a combination of interleukin-8 (IL-8), Eotaxin, and Interferon-γ (IFNγ) was found to be significantly linked to blood oxygenation over seven days. Individually testing the cytokines in receiver operator characteristics analyses identified IL-8 as a strong stratifier for clinical outcomes. Circulating IL-8 dynamics paralleled disease course. We also revealed key transitions in immune transcriptome in patients stratified for circulating IL-8 at three time-points. The study identifies plasma IL-8 as a key pathogenic cytokine linking systemic hyper-inflammation to the clinical outcomes in COVID-19.

## 1. Introduction

The coronavirus disease of 2019, or COVID-19, caused by SARS-CoV-2, caused considerable morbidity and mortality all over the world [1,2,3]. Emerging genetic variants of the virus led to successive waves following the pandemic caused by the original strain in early 2020, even though multiple vaccines have been approved and are in use worldwide [4,5]. To date, more than 675 million individuals have been infected, leading to more than 6.75 million deaths worldwide. Mortal outcomes of COVID-19 mostly lead from acute respiratory distress syndrome (ARDS), which follows systemic hyper-inflammation [3]. Despite a great number of studies aimed at characterizing systemic hyper-inflammation and its link to severe respiratory disease in some patients, the immunopathogenesis of severe COVID-19 remains an enigma [6,7,8,9,10,11,12].

Systemic immunosuppression using corticosteroids remains the most successful therapy [13,14]. Therapies targeting cytokine interleukin-6 are also effective in a group of patients but with heterogeneous efficacy [15,16,17]. Anti-viral therapies, using small molecules or a combination of antibodies, also have very limited efficacy in terms of the number of responding patients [18,19,20,21]. Thus, meticulous characterization of systemic hyper-inflammation in order to identify key molecular targets that are more universal is of major importance. Moreover, biomarkers linked to the hyper-inflammatory response, which parallel the disease course with dependable predictive value for clinical outcomes, are also very useful.

Systemic hyper-inflammation in severe COVID-19 patients features a systemic deluge of cytokines [6,7]. In this study, we performed exploratory analysis on multiplex plasma cytokine data and associated clinical meta-data from a cohort of severe COVID-19 patients, originally recruited in an already completed and reported randomized control trial (RCT) on convalescent plasma therapy (CPT) [22]. We aimed to identify cytokine/s with circulating dynamics closely linked to a quantitative clinical parameter over a significant period of time. Quantitation of the extent of blood oxygenation was represented by the kinetics of the ratio between capillary blood oxygen saturation (SpO2) and the fraction of oxygen delivered in the inhaled air (FiO2), as previously described [12,22]. We employed a machine learning algorithm to shrink a multiple regression model to derive 7-day blood oxygenation from day 1 plasma cytokine levels. We identified plasma interleukin-8 (IL-8) to be a strong predictor of poor clinical outcomes, as patients with higher IL-8 registered less favorable clinical outcomes. The immune transcriptome was also found to be distinctive between patients with higher and lower plasma IL-8 at different time-points. Finally, the kinetics of the plasma level of IL-8 over seven days post-sampling closely paralleled the disease course and predicted outcomes.

## 2. Materials and Methods

### 2.1. Plasma Cytokine Analysis

The plasma cytokine panel dataset used in these exploratory analyses was originally generated from cryo-stored plasma samples, collected at three time-points over seven days post-recruitment, from a cohort of severe COVID-19 patients with ARDS (N = 77), originally recruited for an already completed and reported randomized control trial on convalescent plasma therapy (CTRI/2020/05/025209) at the ID and BG Hospital, Kolkata [22]. The study was approved by the institutional ethics committees of ID & BG Hospital (IDBGH/Ethics/2429) and CSIR-Indian Institute of Chemical Biology, Kolkata, India, in accordance with the Helsinki Declaration. Out of 48 analytes assayed, 36 cytokines, which were detectable in 70% of severe COVID-19 patients in our cohort, were used for analyses [22]. Threshold cycle values from the real-time PCR done for SARS-CoV-2 targets on RNA derived from a nasopharyngeal swab on the day of recruitment (time-point 1 or T1) and data on percent neutralization by plasma in a surrogate virus neutralization assay based on recombinant ACE2 and viral receptor binding domain (RBD) interaction, representing neutralizing antibody content of the plasma, were also used [22].

### 2.2. Machine Learning for Modeling

In order to derive a quantitative parameter for blood oxygenation level, we used the area under curve values for SpO2/FiO2 ratio kinetics for individual patients over seven days following recruitment (SFR_7d_AUC), as described earlier [12,22]. Then, to model this blood oxygenation status based on the circulating cytokine level dataset using a machine learning algorithm, we chose Bayesian information criterion in R software (codes are provided in the Appendix A). As the cytokines data were collected in several batches, before running the multiple regression model, the cytokine dataset was batch-corrected using the ComBat algorithm [23,24].

### 2.3. Receiver Operating Characteristic Curve

The receiver operating characteristic (ROC) curve was prepared using the ‘pROC’ package (Version 1.17.0.1) in R software. The area under curve (AUC) value and cut-off value were determined using the ‘cutpointr’ function. All relevant codes are provided in the Appendix A.

### 2.4. Single-Cell RNA Sequencing Data Analysis

Seurat R software package (Version 4.0.3) was used to analyze two single-cell RNA sequencing data available in GEO Datasets (Accession no. GSE163668 and GSE145926). Dimensionality reduction of the scaled data was performed by using principal component analysis. Finally, TSNE plots were obtained to visualize the expression of CXCL8 and CXCR1 expression in different clusters and among different major immune cells. All relevant codes are provided in the Appendix A.

### 2.5. RNA-Sequencing

The RNASeq library was prepared using Illumina TruSeq Stranded Total RNA Library Prep Kit. Libraries were sequenced on the Illumina NextSeq 550 platform, using high output Kit v2.5 (300 Cycles), at a final library concentration of 1.6 pM. Filtered fastq files were processed using Salmon v1.4.0 [25]. Reference transcriptome Ensembl GRCh38 (release 103) was used for indexing and quantification of genes. TPM values were analyzed using MeV software [26]. Limma tool [27] was used to list differentially expressed genes (DEGs) between patient sub-groups. The list of DEGs (transcripts) (*p* ≤ 0.05) provided by Limma was divided into two groups—(1) upregulated (having all differentially expressed transcripts upregulated) and (2) downregulated (having all differentially expressed transcripts downregulated) before being entered into the online NetworkAnalyst software [28] to obtain the list of enriched pathways (*p* ≤ 0.05) from the Reactome database, separately for upregulated and downregulated genes, as well as the specific genes implicated in these pathways.

### 2.6. Statistics

Kaplan–Meier analyses were performed on Graphpad Prism software or in R. The statistical tests performed are depicted while describing the respective figures.

## 3. Results

### 3.1. Machine Learning to Derive a Minimal Model Based on Plasma Cytokine Levels to Predict Clinical Outcome in Severe COVID-19

Systemic cytokine deluge associated with the hyper-inflammatory state encountered in severe COVID-19 is established as the therapeutic target, and anti-inflammatory corticosteroids have been by far the most successful pharmacotherapy. Targeting individual cytokines such as IL-6 has also been productive, but only in patient subsets [15,16,17]. Targeting cytokines, either as disease biomarkers or therapeutic targets, is limited by the well-documented heterogeneity in the composition of the cytokine deluge in severe COVID-19 [7,22]. To identify cytokines with the most generalized influence on COVID-19 disease outcomes, we attempted a retrospective analysis, powered by machine learning algorithms, on clinical metadata and plasma cytokine concentration data derived from a concluded RCT on CPT, performed in a small Indian cohort of severe COVID-19 patients [22]. The cohort characteristics are depicted in Appendix A. The design of the study is depicted in Figure 1A. Plasma levels of 36 different cytokines (sampled at ARDS diagnosis, <7 days post-hospitalization) were taken as features to describe the tissue response of the patients in terms of blood oxygenation over 7 days post-sampling, taking the area under curve for the kinetics of SpO2/FiO2 ratio as a quantitative surrogate (termed SFR_7d_AUC). The higher the SFR_7d_AUC value, the better the blood oxygenation over 7 days [22]. Figure 1B shows the heterogeneity of cytokines clusters based on plasma concentrations in different patients with varying levels of SFR_7d_AUC.

We conducted a multivariate regression analysis to express SFR_7d_AUC as a function of thirty-six variables represented by the plasma concentrations of individual cytokines. For optimal model shrinkage, we performed a widely used machine learning algorithm, viz. Bayesian information criterion or BIC (Figure 2A, Appendix A). Three variables (or cytokines) were featured in BIC viz. the proinflammatory cytokine IL-8, the effector T cell-derived cytokine interferon-γ (IFNγ), and the chemokine Eotaxin related in the following equation: SFR_7d_AUC ~ Y = 3.828 × [Eotaxin] + 17.122 × [IFNγ] − 21.334 × [IL-8] + 673.188

Values of the function (Y) calculated for each patient, but excluding the intercept value (designated as BIC-Y), were plotted against the actual SFR_7d_AUC values, which showed a strong linear correlation, as expected (Figure 2B). As described earlier, SFR_7d_AUC is a quantitative surrogate for blood oxygenation and, thus, also for the extent of involvement/resolution of pulmonary tissue pathology in COVID-19. We hypothesized that the BIC-Y value could also have the potential to predict final disease outcomes. On performing ROC analysis, for predicting mortal outcome, we determined a BIC-Y cut-off value of 275.0964, although with very low specificity (Figure 2C, Appendix A). Nevertheless, Kaplan–Meier curve analyses revealed that patients scoring a higher than cut-off BIC-Y value registered a significant survival benefit with a Mantel Haenszel hazard ratio of 0.2958, *p* = 0.0059 in the Mantel–Cox log-rank test (Figure 2D). Patients with higher BIC-Y values also were found to have earlier mitigation of hypoxia (Appendix A) as well as earlier remission, as noted from time to discharge from the hospital (Figure 2E).

### 3.2. Predictive Value of Individual Cytokines Featured in BIC-Derived Regression Model and Identification of IL-8 as an Independent Predictor

As the sensitivity and specificity profile of BIC-Y was not robust, and because plasma levels of individual cytokines would have much greater clinical applicability as a prognostic biomarker, instead of the value computed from a regression analysis, we next performed ROC analyses for predicting fatal outcomes, using plasma concentrations of Eotaxin, IFNγ, and IL-8 individually. We found that the AUC value of the ROC curve was highest for IL-8 (Figure 3A), compared to both Eotaxin (Appendix A) and IFNγ (Appendix A).

When we stratified patients in our cohort as having higher or lower plasma levels than computed cut-offs for either Eotaxin or IFNγ, none of the subgroups showed any comparative benefit either in terms of survival (Appendix A) or in terms of earlier remission (Appendix A). However, on stratifying patients based on a computed cut-off value of 7.7903 pg/mL for plasma concentration of IL-8, with a sensitivity of 90.48% and specificity of 48.21%, patients having IL-8 levels lower than the cut-off level (IL8_lo_) registered significantly favorable final outcome in terms of 30-day survival on Kaplan–Meier curve analysis with a hazard ratio of 0.3069, *p*-value 0.0077 (Figure 3B, Appendix A). IL8_lo_ patients were also found to have earlier mitigation of hypoxia (Figure 3C) as well as earlier remission (Figure 3D, Appendix A).

Viral load at T1 (represented by CT values from RT-PCR for SARS-CoV-2 targets on nasopharyngeal swab samples) was not significantly different between IL8_lo_ and IL8_hi_ patients (Appendix A). Plasma-neutralizing antibody content at T1 (represented by percent neutralization of ACE2-RBD interaction) was quite similar among the IL8_lo_ and IL8_hi_ patients (Appendix A).

We did not find any considerable effect of age, gender, and major comorbidities on the plasma level of IL-8 on the day of enrolment among the severe COVID-19 patients recruited in the cohort (Appendix A). We also performed a series of sub-class analyses to validate the predictive value of plasma IL-8 concentration, e.g., between male and female ARDS patients, between patients who were diabetic or hypertensive and who were not, and patients who received different therapies, viz. corticosteroids, remdesivir, and convalescent plasma (Figure 3E). While males, non-diabetics, and patients receiving corticosteroids or remdesivir as part of their therapies showed statistically significant survival benefits when they were IL8_lo_, these sub-class analyses were handicapped by lower sample sizes for some of the sub-classes and, thus, warrant further exploration in bigger cohorts of severe COVID-19 patients to establish the observations.

Of note here, in patients who received COVID convalescent plasma (CCP) as part of their therapy (which was after the sampling for IL-8 measurement was performed), initial plasma IL-8 level was not predictive of clinical outcomes. On the other hand, patients not receiving CCP did register this predictive stratification. This may point to the rapid anti-inflammatory effect of CCP reported earlier, which may dissipate the effect of higher levels of circulating IL-8 prior to CCP transfusion [7,22].

### 3.3. Exploring Cellular Sources of IL-8 and IL-8-Responder Cells in Peripheral Blood and Bronchoalveolar Lumen

To glean further insight into cellular sources of IL-8 and the IL-8 responder cells in severe COVID-19 patients, we conducted a series of targeted re-analyses of two publicly available single-cell RNA sequencing datasets, GSE163668 (blood mononuclear cells) and GSE145926 (cells derived from bronchoalveolar lavage), generated from COVID-19 patients [11,29]. As expected from our present data as well as from previously reported data [30], significantly increased expression of both CXCL8 (gene encoding IL8) and its receptor CXCR1 was apparent in both peripheral immune cells as well as cells in bronchoalveolar lavage fluid (BALF) in severe COVID-19, as compared to cells derived from COVID-19 patients with milder disease course (Figure 4A–H). Expression of the other IL-8 receptor CXCR2 was insignificant and, thus, not considered for analyses.

We explored different cell subsets individually, marked by their differential expression of specific cell surface markers commonly used to identify them. In GSE163668, the circulating myeloid origin cells (expressing CD16 or FCGR3A, CD14, CD11c or ITGAX, CD11b or ITGAM, marking monocytes, macrophages, and conventional dendritic cells) were found to have the highest expression of IL-8 (Figure 5A). On the other hand, B (marked by CD19) or T lymphocytes (CD4 and CD8) and natural killer cells (marked by NCAM1) showed negligible expression (Figure 5A). Expression of the IL-8 receptor (CXCR1) was also found to be highest in the myeloid cells circulating in the blood (Figure 5B).

When the scRNAseq data (GSE145926) from BALF cells were analyzed, a significant expression of IL-8 was also found in epithelial cells (marked by EPCAM1) (Figure 5C), apart from the myeloid cells. Expression of the receptor CXCR1 was again restricted to the cells of myeloid origin (Figure 5D). The infected epithelium, thus, may play a key role in the local amplification of the inflammatory cascade by producing IL-8.

### 3.4. Systemic Mechanistic Insight from Transcriptome Studies on Circulating Immune Cells Compared between IL8_hi_ and IL8_lo_ Patients at Different Time-Points

To explore global transcriptome response in circulating immune cells in response to changes in plasma IL-8 across the disease course over seven days following recruitment, we looked at IL-8 levels in plasma at two follow-up time-points, time-point 2 (T2) on the 3rd or 4th day following initial enrolment (T1) and time-point 3 (T3) on the 7th day following T1. We stratified the patients at each time-point (T1, T2, and T3) into IL8_hi_ and IL8_lo_ sub-groups based on the cut-off value derived at T1. RNA sequencing was performed on peripheral blood mononuclear cell samples from 17 selected patients representing the IL8_hi_ and IL8_lo_ sub-groups at each time-point. Accordingly, peripheral blood transcriptome was generated from nine patients falling into the IL8_hi_ group at T1 (designated T1_hi_), eight patients falling into the IL8_lo_ group at T1 (T1_lo_), seven patients falling into the IL8_hi_ group at T2 (T2_hi_) and six patients falling into the IL8_lo_ group at T2 (T2_lo_). Finally, a comparison was performed on five patients who were classified as T1_hi_ and ended up falling into the IL8_lo_ group at time-point T3 (designated T3_lo_). After pre-processing and normalization, differentially regulated genes were compared between these sub-groups (Figure 6A–C).

Major pathways enriched by the upregulated genes expectedly represented the systemic hyper-inflammation, including toll-like receptor activation cytokine signaling, ECM biosynthesis, activation of fibroblasts, and activation of pathways associated with transcription of RNA viruses as well as TLR-mediated type I interferon response (Figure 6D, Appendix A). On the other hand, the major pathways enriched by downregulated genes in the T1_hi_ sub-group in comparison to the T1_lo_ sub-group were mainly transcriptional regulation, host–virus interaction, DNA repair, TCA cycle, and others which represented normative anti-viral responses (Figure 6G, Appendix A). A very similar transcriptional landscape was apparent at T2 as well − T2_hi_ sub-group showing upregulation of pathways linked to hyper-inflammation, viz. TLR and cytokine signaling and downregulation of pathways linked to anti-viral immune response, e.g., lymphocyte activation (Figure 6E,H, Appendix A).

Finally, we looked for changes in the immunocellular transcriptional landscape in patients who, despite being IL8_hi_ at T1 in the course of the disease, had a reduction in their plasma IL-8 levels and were classified as T3_lo_. Interestingly, compared to T1, at T3, both the upregulated and downregulated pathways were representative of tissue regeneration response, mitigation of inflammation, and dissipation of anti-viral response, indicating a transition to the state of disease remission (Figure 6F,I, Appendix A).

### 3.5. Circulating IL-8 Dynamics Parallels Disease Course

Then, to further explore how patients would fair in response to changes in plasma IL-8 concentrations along the disease course, we assessed final disease outcomes in patients falling into the IL8_hi_ and IL8_lo_ groups at each time-point (cut-off value as per T1, represented in data shown in Figure 2A–D). It was evident that most patients who remained in the IL8lo group at all three time-points or patients in the IL8_hi_ group at T1 but became IL8_lo_ at T2 or T3, met with disease remission. On the other hand, a large majority of patients remaining in the IL8_h_i group all along their disease course had a fatal outcome (Figure 7A).

To further validate this aspect, we performed ROC analyses on plasma IL-8 levels for T2 and T3 as well. Computed cut-off values were not much changed (7.7903 pg/mL at T1, 9.867 pg/mL at T2, and 10.4614 pg/mL at T3), while the specificity of these cut-off levels increased with time, reaching 87.5% at T3 (Figure 7B). Indeed, on stratifying the patients based on a cut-off value of 9.85 pg/mL, which is the mean cut-off value for all three time-points, again, the resulting IL8lo group patients (as per plasma level at T1) registered significantly favorable outcomes both in terms of survival as well as time to remission (Appendix A). Thus, a high level of circulating IL-8 was linked to unfavorable disease outcomes, and its modulation over 7 days paralleled the disease course toward remission or non-remission. The specificity of the computed cut-off plasma level, thus, increased with days along the disease course.

## 4. Discussion

Systemic hyper-inflammation in COVID-19 characteristically follows the initial mildly symptomatic phase of infection with SARS-CoV-2 [1,2,3]. Pathological outcomes of progressively deficient blood oxygenation function of the lungs are concomitant to the hyper-inflammatory phase [31]. Gradual decommissioning of alveoli has been thought to be due to myriad micro-pathologies [31,32]. An expanded myeloid cell compartment in the periphery is reflected in myeloid infiltration in the lung beds [9,10,11]. The systemic deluge of cytokines is projected to play a major role in tissue damage as well as infiltration of the inflammatory cells. In addition, a hypercoagulable state in the microvasculature is also projected based on evidence for pulmonary microthrombotic events [33,34]. This also contributes to alveolar decommissioning. A plausible mechanistic link of the microvascular hypercoagulable state with the systemic hyper-inflammation may lie with the expanded myeloid cells, which express critical coagulation regulators, viz. suPAR, both in circulation and in tissue [12].

A number of studies have looked into the cytokine deluge and its predictive link with the clinical outcomes in cohorts from different parts of the world [7,35,36,37,38,39,40,41,42]. Here, in an Indian cohort of severe COVID-19 patients, we first utilized a machine-learning algorithm to derive a multiple regression model that could select a minimal combination out of 36 different cytokines (sampled <7 days post-hospitalization), which could act as features to describe the disease progression. We utilized a quantitative surrogate for the disease progression, SFR_7d_AUC, to be used as the function modeled by the features. The Bayesian information criterion algorithm was used to derive the minimal model [43]. This led us to a model based on plasma levels of IL-8, IFNγ, and Eotaxin which fitted the SFR_7d_AUC data. An index based on plasma levels of these three cytokines early in the disease course and derived from this model, which we termed BIC-Y, was found to have a rather weak predictive value for final clinical outcomes.

On examining the predictive value of the three individual cytokines, we found that only IL-8 had the ability to stratify patients based on final clinical outcomes. Moreover, follow-up sampling of plasma at two time-points over 7 days following recruitment revealed that the circulating IL-8 dynamics closely paralleled the disease course. Re-analyses of single-cell RNAseq datasets from other studies revealed that the expanded myeloid compartment largely contributed to IL-8 production, although lung epithelial cells also contributed. Another previous study also implicated the local abundance of IL-8 in the lung in contributing to disease severity in COVID-19 [39]. Patient subgroups stratified based on cut-off level of IL-8 in their plasma also showed differential immunocellular transcriptome, which represented the systemic hyper-inflammatory state in the IL8_hi_ patient subgroup, which was mitigated as the plasma level of IL-8 was reduced in the course of the disease. IL-8 has also been identified in a few other studies to be associated with clinical outcomes [35,36,37,38,39,40,41]. Our study complements these earlier reports by showing the importance of the longer-term dynamics of IL-8 and its link with disease outcomes. Further validation of these data in other cohorts will be useful in establishing IL-8 as a dependable biomarker in severe COVID-19 as well as a potential therapeutic target.

## Figures and Tables

**Figure 1 viruses-15-00549-f001:**
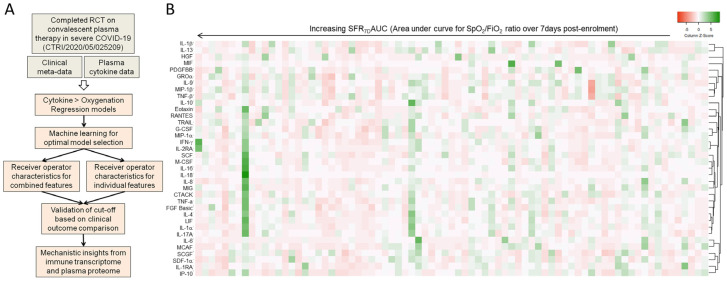
Circulating cytokines and blood oxygenation in severe COVID-19 patients. (**A**) Experimental schema for the study. (**B**) Correlation (Pearson) clustering of plasma abundance of cytokines shown in order of increasing blood oxygenation (represented by area under curve of the SpO2/FiO2 ratio kinetics over 7 days following recruitment.

**Figure 2 viruses-15-00549-f002:**
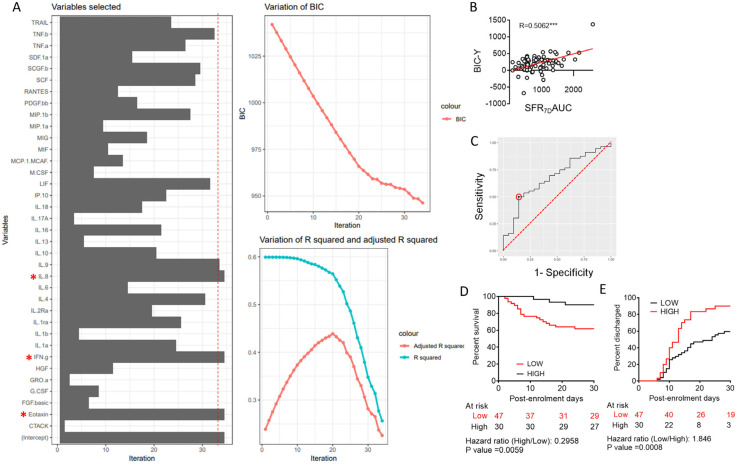
Machine learning to derive a minimal predictive model to describe blood oxygenation using plasma cytokine abundance in severe COVID-19 patients. (**A**) Bayesian information criterion applied to multiple regression model to derive SFR_7d_AUC based on plasma concentration of cytokines at time-point 1 (day of recruitment). The minimization of BIC value and evolution of R2 values are shown in the right panel with elimination of cytokines to derive a minimally effective model. Red * indicates the cytokines derived from the analysis. (**B**) Correlation of the three cytokine BIC-Y index with actual SFR_7d_AUC values in severe COVID-19 patients. Pearson’s R-value is shown, *** *p* < 0.0005. (**C**) Receiver operator characteristic curve (black line) of BIC-Y values to derive remission versus non-remission (death); the marked point denotes the specificity and sensitivity with the derived cut-off. (**D**) Survival of patients until day 30 post-enrolment is compared in a Kaplan–Meier curve between patients with BIC-Y values above (black line) or below (red line) the cut-off derived from the ROC curve shown in (**C**). Surviving patients were censored on day 30 post-enrolment. (**E**) Hospital stay duration (time to remission) of the patients from both groups (BIC-Y values above or below cut-off) since the day of enrolment are plotted in an ascending Kaplan–Meier curve. Deaths and non-remission at day 30 post-enrolment were censored. For the outcome comparisons shown in (**D**,**E**), the Mantel–Cox log-rank test was performed. The number of patients at risk on different days and the Mantel–Haenszel hazard ratio is shown.

**Figure 3 viruses-15-00549-f003:**
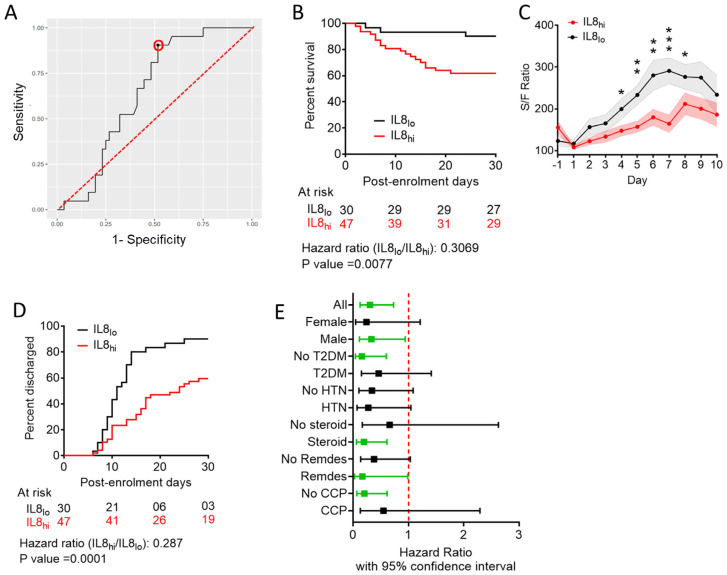
Plasma level of IL-8 and clinical outcomes in severe COVID-19 patients. (**A**) Receiver operator characteristic curve (black line) of plasma levels of IL-8 to derive remission versus non-remission (death), the marked point denotes the specificity and sensitivity with the derived cut-off (red circle). The red dotted line denotes the diagonal. (**B**) Survival of patients till day 30 post-enrolment is compared in a Kaplan–Meier curve between patients with plasma level of IL-8 below (IL8_lo_, black line) or above (IL8_hi_, red line) the cut-off derived from the ROC curve shown in (**A**). Surviving patients were censored on day 30 post-enrolment. (**C**) Comparison of SpO2/FiO2 ratio kinetics over 7 days following plasma sampling between IL8_lo_ (black line) and IL8_hi_ (red line) subgroup of patients. * *p* < 0.05, ** *p* < 0.005, *** *p* < 0.0005 from unpaired *t*-tests. (**D**) Hospital stay duration (time to remission) of the patients from both groups, IL8_lo_ (black line) and IL8_hi_ (red line), since the day of enrolment is plotted in an ascending Kaplan–Meier curve. Deaths and non-remission at day 30 post-enrolment were censored. For the outcome comparisons shown in (**B**,**D**) Mantel–Cox log-rank test was performed. Number of patients at risk on different days and the Mantel–Haenszel hazard ratio is shown. (**E**) Mantel–Haenszel hazard ratio for Kaplan–Meier analysis for survival in subgroups of patients, for IL8_lo_ patients, compared with IL8_hi_ patients. Analyses showing Mantel–Cox log-rank test *p*-values <0.05 are indicated in green.

**Figure 4 viruses-15-00549-f004:**
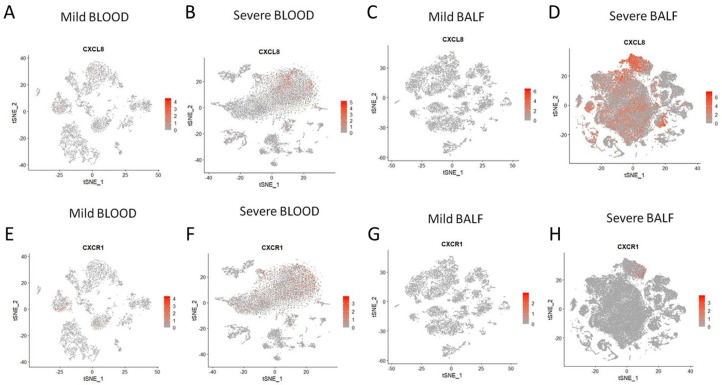
Re-analyses of single-cell RNA sequencing data to show expression of IL-8 and its receptor among all cells. (**A**–**D**) Abundance of IL-8 transcript (CXCL8) among different subsets of cells isolated from peripheral blood (**A**,**B**) and bronchoalveolar lavage fluid (**C**,**D**), compared between mild (**A**,**C**) and severe (**B**,**D**) COVID-19 patients. (**E**–**H**) Abundance of transcript for IL-8 receptor (CXCR1) among different subsets of cells isolated from peripheral blood (**E**,**F**) and bronchoalveolar lavage fluid (**G**,**H**), compared between mild (**E**,**G**) and severe (**F**,**H**) COVID-19 patients. Data analyzed from public datasets GSE163668 (blood cells) and GSE145926 (cells from bronchoalveolar lavage).

**Figure 5 viruses-15-00549-f005:**
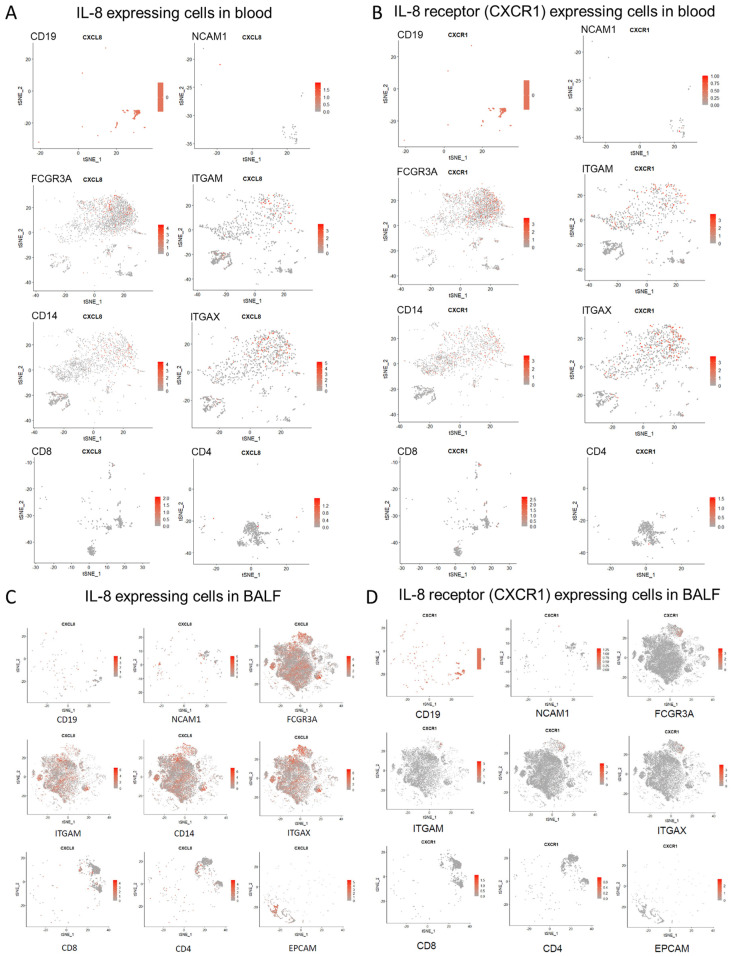
Re-analyses of single-cell RNA sequencing data to show individual immune cells expressing IL-8 and its receptor in severe COVID-19 patients. (**A**) Abundance of IL-8 transcript (CXCL8) among different cell subsets from peripheral blood of severe COVID-19 patients, defined by expression of characteristic transcripts. (**B**) Abundance of the transcript for IL-8 receptor (CXCR1) among different cell subsets from peripheral blood of severe COVID-19 patients, defined by expression of characteristic transcripts. Data analyzed from public dataset GSE163668. (**C**) Abundance of IL-8 transcript (CXCL8) among different cell subsets from bronchoalveolar lavage of severe COVID-19 patients, defined by expression of characteristic transcripts. (**D**) Abundance of the transcript for IL-8 receptor (CXCR1) among different cell subsets from bronchoalveolar lavage of severe COVID-19 patients, defined by expression of characteristic transcripts. Data analyzed from public dataset GSE145926.

**Figure 6 viruses-15-00549-f006:**
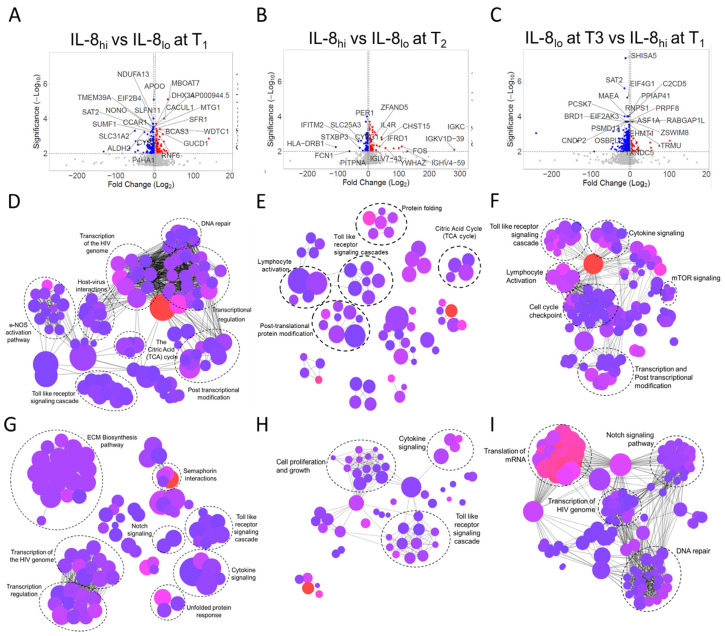
Analyses of immunocellular transcriptome from peripheral blood mononuclear cells from severe COVID-19 patients. (**A**–**C**) Differentially expressed genes (more than 2-fold change) shown in a volcano plot, analyzed from RNA sequencing data generated from peripheral blood mononuclear cells, compared between patients falling into the IL8_hi_ group at time-point T1 (N = 9), designated T1_hi,_ and patients falling into the IL8_lo_ group at time-point T1 (N = 8), designated T1_lo_ (**A**), between patients falling into the IL8_hi_ group at time-point T2 (N = 7), designated T2_hi,_ and patients falling into the IL8_lo_ group at time-point T2 (N = 6), designated T2_lo,_ (**B**) and, finally, between patients who were classified as IL8_hi_ at time-point T1 (N = 5), designated T1_hi,_ and ended up falling into the IL8_lo_ group at time-point T3, designated T3_lo_ (**C**). (**D**–**F**) Network depictions of major pathway groups enriched by genes upregulated in T1_hi_ compared to T1_lo_ (**D**), upregulated in T2_hi_ compared to T2_lo_ (**E**), and upregulated in T3_lo_ compared to T1_hi_ (**F**). (**G**–**I**) Network depictions of major pathway families enriched by genes downregulated in T1_hi_ compared to T1_lo_ (**G**), downregulated in T2_hi_ compared to T2_lo_ (**H**), and downregulated in T3_lo_ compared to T1_hi_ (**I**).

**Figure 7 viruses-15-00549-f007:**
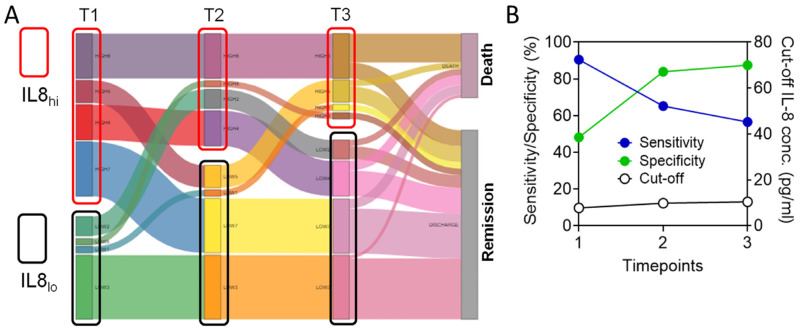
Plasma IL-8 dynamics parallel disease course and stratify final outcomes in severe COVID-19. (**A**) Sankey chart showing subgroups of severe COVID-19 patients variably falling into either IL8_hi_ or IL8_lo_ groups at three different time-points, stratified in terms of final outcomes. (**B**) Plot shows data from receiver operator characteristics analyses for plasma IL-8 levels at three time-points in terms of final fatal outcomes, showing the respective cut-off values (black line, right Y axis represented as percent) as well as sensitivity (blue line) and specificity (green line) for those cut-off values (left Y axis represented as plasma concentration of IL-8).

## Data Availability

Data are available from the corresponding author. The RNA sequencing data will be uploaded to a public database before publication.

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
