# Peer review of "Circulating Interleukin-8 Dynamics Parallels Disease Course and Is Linked to Clinical Outcomes in Severe COVID-19"

_viruses, 2023, doi:10.3390/v15020549_

Round 1

Reviewer 1 Report

I found this manuscript quite interesting. Results are very well described. However, I find minor questions and some comments before acceptance.

-Methods: The sample size was 77 ARDS, was this number enough in order to assume the results? Any statistical power to obtain the minimum sample size was made?

-Results: I find necessary a table 1 (even in suppl files) to characterize the sample. Age, sex, comorbidities are important. Differences in those parameters could produce bias. The results of IL-8 should be performing in a multivariate analyses including the corresponding characteristics.

-Discussion: This part is poor. It has to be improved. The authors should compare their results with other works, focus on other similar studies trying to identify cytokines and severity (doi: 10.3390/jcm10092017, doi: 10.1016/j.cyto.2022.155804) and the importance of IL-8 (doi: 10.3390/cells11182912, doi: 10.1016/j.chest.2022.06.007). I recommend those articles.

Author Response

File attached

Reviewer 2 Report

This is an interesting and comprehensive analysis suggesting that circulating IL-8 parallel disease outcome/severity. Overall, the manuscript is rigorous but a few minor points need to be addressed before publication.

1) The x-axis in Figure 3E should be displayed on a logarithmic scale.

2) HRs ideally need to be adjusted for outcome related confounders.

3) Time point data in Section 3.4 should be handled using a repeated measures, mixed effects model.

4) Given the numerous statistical results reported, multiplicity correction is advised.

5) Study limitations need to be addressed in the discussion section.

6) English language editing is recommended.

Author Response

File attached
